# Learning Reliable Logical Rules with SATNet

**Zhaoyu Li**[1,2]**, Jinpei Guo**[4]**, Yuhe Jiang**[1]**, Xujie Si**[1,2,3]
[1]University of Toronto, [2]Vector Institute, [3]Mila, [4]Shanghai Jiao Tong Univeristy
{zhaoyu, six}@cs.toronto.edu

## Abstract

Bridging logical reasoning and deep learning is crucial for advanced AI systems. In this work, we present a new framework that addresses this goal by generating interpretable and verifiable logical rules through differentiable learning, without relying on pre-specified logical structures. Our approach builds upon SATNet, a differentiable MaxSAT solver that learns the underlying rules from input-output examples. Despite its efficacy, the learned weights in SATNet are not straight-forwardly interpretable, failing to produce human-readable rules. To address this, we propose a novel specification method called "maximum equality", which enables the interchangeability between the learned weights of SATNet and a set of propositional logical rules in weighted MaxSAT form. With the decoded weighted MaxSAT formula, we further introduce several effective verification techniques to validate it against the ground truth rules. Experiments on stream transformations and Sudoku problems show that our decoded rules are highly reliable: using exact solvers on them could achieve 100% accuracy, whereas the original SATNet fails to give correct solutions in many cases. Furthermore, we formally verify that our decoded logical rules are functionally equivalent to the ground truth ones.

## 1 Introduction

Logical reasoning is a fundamental cognitive skill that empowers humans to organize knowledge in a structured form, reason about relationships between different pieces of information, and draw inferences based on logical rules and constraints. In recent years, there has been a growing research interest in integrating advanced deep learning systems with the ability of logical reasoning [9, 10, 23, 32]. A notable outcome in this field is SATNet [32], a differentiable MaxSAT solver based on a low-rank semidefinite programming (SDP) approach. SATNet utilizes a learnable matrix $S$ to capture the underlying structure of logical rules and dynamically adjusts its weight from input-output examples. This innovative design enables SATNet to learn sophisticated logical rules, such as the parity function and Sudoku puzzles.

Despite the promising results of SATNet, the interpretability of its learned weights remains unclear. This limitation arises from the fact that the clause matrix $S$ learned by SATNet is not ternary[1], preventing its explicit interpretation as a set of logical rules as originally motivated [33]. Consequently, this poses a challenge to the wider adoption of SATNet in real-world applications that require transparent decision-making processes.

To address this limitation, we present a new framework that extracts a set of interpretable logical rules from the learned weights in SATNet. We first demonstrate that merely rounding $S$ to a ternary matrix is insufficient for interpretation and then show empirical and theoretical evidence that $S$ may fail to capture the underlying logical rules in certain cases. Instead of directly decoding $S$, we propose utilizing the matrix $C = S^T S$ to derive explicit logical rules, based on the SDP formulation in

---

[1]In SATNet's original formulation, it assumes the $S$ matrix can only consist of $1$, $-1$, and $0$ elements to represent the logical formula in MaxSAT form. However, after training, $S$ is continuous rather than ternary.

37th Conference on Neural Information Processing Systems (NeurIPS 2023).

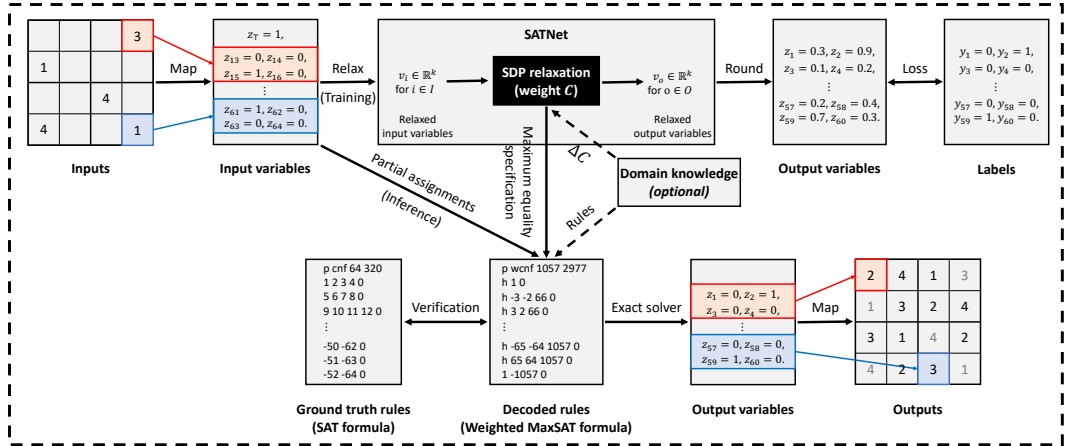

Figure 1: An example pipeline of our framework for $4 \times 4$ Sudoku problem. We first train SATNet to learn the Sudoku rules from data. After training, we use maximum equality specification to decode a set of interpretable rules as a weighted MaxSAT formula from the learned weight $C$. Given the decoded rules, we call an exact solver for inference and further verify them with ground truth rules. Optional domain knowledge can be incorporated through numerical weights or propositional rules.

**SATNet.** Specifically, we introduce a novel specification formulation called "maximum equality", which transforms the learned matrix $C$ into a weighted MaxSAT formula. We rigorously prove that our proposed formulation has a powerful expressive ability, enabling $C$ to represent any propositional logical rules under proper parameterization.

With the decoded logical rules, we can employ an exact solver (e.g., Gurobi [13]) to replace SATNet during inference, rendering the solving process more accurate and transparent. Additionally, our specification method offers flexibility in incorporating prior domain knowledge in propositional logic. This integration can be done through either the learned weights $C$ or the weighted MaxSAT formula, bolstering the reliability of our learned or decoded rules. To further validate the correctness of the decoded logical rules, we propose several verification techniques that prove the equivalence between our decoded weighted MaxSAT formula and the human-written rules in SAT form. In particular, we establish that if the optimal solution of the weighted MaxSAT formula always corresponds to a satisfying assignment of the SAT formula for any valid partial inputs, then these two formulas function equivalently. Notably, when the space of inputs becomes excessively huge, we introduce a sufficient condition for an efficient equivalence check, which assures not only the functional equivalence but also a similar distribution of solution space. We summarize our proposed framework in Figure 1.

To evaluate the efficacy of our proposed framework, we conduct a series of experiments on various tasks, including stream transformations (i.e., the parity function, addition, and counting) and Sudoku puzzles. For stream transformations, we strictly verify that our decoded weighted MaxSAT formula is functionally equivalent to the human-written logical rules, which ensures that it accurately captures the underlying logic of these tasks. On Sudoku puzzles, we show that using exact solvers on our generated formula could achieve 100% accuracy on the testing set and further verify that it functions equivalently to the ground truth for all unique puzzle inputs. Conversely, SATNet fails to achieve perfect accuracy on Sudoku puzzles, where we certify this is because SATNet gives a sub-optimal solution to our decoded weighted MaxSAT formula.

In summary, our contributions include:

- We pinpoint the interpretability challenges inherent in the original SATNet and present both empirical and theoretical insights into SATNet's inability to capture logical rules in some cases.

- We introduce a novel interpretation method that decodes a set of logical rules as a weighted MaxSAT formula from SATNet and enables the flexible integration of domain knowledge.

- We propose multiple verification approaches to validate the correctness of our decoded rules with human-written ground truth rules.

- Experiments on stream transformations and Sudoku puzzles show that our decoded weighted MaxSAT formula exactly captures the underlying logical rules and is verifiably reliable.

## 2 Background

### 2.1 Problem Formulation

In this paper, we focus on problems whose inherent logical rules can be expressed using propositional logic. For a given logical problem, we define a set of $n_d$ problem Boolean variables $z_1, \ldots, z_{n_d}$ to represent the potential relationships within the problem and express the ground truth logical rules as a SAT formula $\phi(z_1, \ldots, z_{n_d})$. The dataset $\mathcal{D}$ consists of input-output examples $(\mathbf{x_i}, \mathbf{y_i})$, where each example partitions $n_d$ variables to an input set $Z_{\mathcal{I}}$ and an output set $Z_{\mathcal{O}}$ and assigns a value to each variable accordingly. The task is to learn the underlying logical rules from the datasets $\mathcal{D}$ such that the learned rules $\phi'$ can accurately map the input variables $z_i \in Z_{\mathcal{I}}$ to the output variables $z_o \in Z_{\mathcal{O}}$, similar to how the ground truth rules $\phi$ conduct the mapping.

Let's take $4 \times 4$ Sudoku as an example. In this case, we have a total of 64 variables, where each successive group of 4 variables represents a single grid in the puzzle that can hold values ranging from 1 to 4. Each input $\mathbf{x_i}$ in the dataset $\mathcal{D}$ corresponds to the initially provided values in a Sudoku puzzle, and the output $\mathbf{y_i}$ represents the solved values of that puzzle. We transform each input-output example to construct an input set $z_i \in Z_{\mathcal{I}}$ and an output set $z_o \in Z_{\mathcal{O}}$. For instance, if variables $z_1, z_2, z_3, z_4$ correspond to the values of the first grid in the Sudoku board, and the given value is 4, then variables $z_1, \ldots, z_4 \in Z_{\mathcal{I}}$ with the configuration $z_1 = 0, z_2 = 0, z_3 = 0, z_4 = 1$. The task is to learn the inherent Sudoku rules from the dataset $\mathcal{D}$ such that given the initial known values of a Sudoku puzzle, the learned logical rules can accurately determine the values of the remaining cells.

### 2.2 SATNet

SATNet [32] is a differentiable MaxSAT solver that models the underlying rules $\phi$ in the data as a MaxSAT problem and solves it using SDP relaxation. It assumes that the logical rules are written in conjunctive normal form with $n_d$ defined variables $\tilde{Z} \in \{-1, 1\}^{n_d}$ and $m$ clauses, which can be represented by a clause matrix $\tilde{S} \in \{1, -1, 0\}^{m \times n_d}$. Specifically, each element $\tilde{s}_{ij}$ in $\tilde{S}$ denotes the sign of variable $\tilde{z}_i$ in clause $j$. The MaxSAT problem can be formulated as follows:

$$\underset{\tilde{Z} \in \{-1,1\}^{n_d}}{\text{maximize}} \sum_{j=1}^{m} \bigvee_{i=1}^{n_d} \mathbf{1}\{\tilde{s}_{ij}\tilde{z}_i > 0\}. \tag{1}$$

SATNet applies semidefinite relaxation to Equation 1 by relaxing each variable $\tilde{z}_i$ to a unit vector $v_i \in \mathbb{R}^k$ with respect to a unit "truth direction" $v_\top$ based on the randomized rounding [12]. It additionally introduces a coefficient vector $s_\top = \{-1\}^m$ associated with the truth vector and writes the SDP relaxation of the MaxSAT problem as:

$$\underset{V \in \mathbb{R}^{k \times (n_d+1)}}{\text{minimize}} \langle S^T S, V^T V \rangle, \quad \text{s.t. } \|v_i\| = 1, i = \{\top, 1, \ldots, n_d\}, \tag{2}$$

where $V = [v_\top, v_1, \ldots, v_{n_d}] \in \mathbb{R}^{k \times (n_d+1)}$ and $S = [s_\top, \tilde{s}_1, \ldots, \tilde{s}_n] \in \mathbb{R}^{m \times (n+1)^2}$.

Given the input variables $z_i \in Z_{\mathcal{I}}$, SATNet relaxes their values to unit vectors and solves Equation 2 by the mixing method [31], a fast coordinate descent approach. In this method, each output vector $v_o$ is updated iteratively as follows:

$$v_o = -g_o / \|g_o\|, \text{ where } g_o = V S^T s_o - \|s_o\|^2 v_o, \text{ for } o \in \mathcal{O}. \tag{3}$$

The updates in Equation 3 are proven to converge to the globally optimal fixed point of the SDP [31]. After obtaining the relaxed outputs $v_o$ from coordinate descent, SATNet converts each output variable vector to a probabilistic value by $p(z_o = 1) = \cos^{-1}\left(-v_i^T v_\top\right)/\pi$ and computes the binary cross-entropy loss between the ground-truth variable assignments $z_o$. When back-propagating the loss, SATNet employs a custom algorithm that uses an efficient coordinate descent algorithm to directly compute the gradient of $S$, without explicitly unrolling the forward-pass computations. Such features enable SATNet to learn the $S$ matrix efficiently from input-output examples.

It is worth noting that SATNet requires to specify the number of clauses $m$ before learning a clause matrix $S$. Moreover, to improve the representational power of $S$, SATNet further incorporates $n_a$ auxiliary variables alongside the $n_d$ problem-defined variables. These auxiliary variables are considered output variables but are not involved in the loss computation.

---

[2]In the original paper of SATNet [32], each row of $S$ is also normalized using the constant $\sqrt{4 \sum_j |\tilde{s}_{ij}|}$. However, in cases where $S$ is learned (rather than provided), this normalization can be omitted for simplicity.

# 3 Methodology

In this section, we first explore the issue of uninterpretability in SATNet by analyzing the clause matrix $S$. We then revisit the formulation of SATNet and present our specification approach that decodes a set of interpretable logical rules from the learned weights $C = S^T S$. After this, we show that our method also allows for the flexible incorporation of domain knowledge via additional rules in propositional logic or the additive weights of $C$. To validate the correctness of our decoded rules, we introduce several verification strategies to compare them with the human-written ones.

## 3.1 Uninterpretability of SATNet

SATNet utilizes the clause matrix $S$ to capture the underlying structure of logical rules and learns its weight through input-output examples. However, SATNet doesn't enforce $S$ to be ternary during training, making it hard to directly obtain a set of interpretable logical rules from $S$. Towards deriving human-readable rules from SATNet, we first conduct a range of experiments on the $S$ matrix[3].

**Rounding $S$ to a ternary matrix.**   One straightforward idea to derive a ternary matrix from $S$ is to round its weights to the values $\{1, -1, 0\}$. To test this approach, we first train SATNet on the $4 \times 4$ Sudoku dataset, which achieves 100% training accuracy and 99.89% testing accuracy. Subsequently, we apply a threshold function $f_{\text{thre}}$ to round each entry of $S$ with a threshold value $\epsilon > 0$:

$$f_{\text{thre}}(s_{ij}) = \begin{cases} s_{ij}/|s_{ij}|, & \text{if } |s_{ij}| > \epsilon, \\ 0, & \text{otherwise.} \end{cases}$$

In our experiments, we exhaustively evaluate all possible threshold values and round $S$ to a set of ternary matrices. We assess the correctness of these matrices by solving Sudoku test instances using an exact MaxSAT solver applied to the corresponding MaxSAT formulas. Our empirical result shows that none of these ternary matrices could yield the correct logical rules, as they fail to solve *any* Sudoku puzzles in the testing set. This finding demonstrates that simply rounding $S$ is insufficient for interpreting the learned weights in SATNet.

**Substituting ground truth rules into $S$.**   To investigate the representational capability of the clause matrix $S$, we conduct an experiment by injecting human-written ground truth rules of $4 \times 4$ Sudoku puzzles into $S$. If $S$ is truly capable of capturing such logical rules, we would expect SATNet to solve Sudoku test instances quite well when provided with the ground truth rules. Contrarily, SATNet achieves only 53.95% solving accuracy on the testing dataset, giving incorrect solutions for a substantial number of Sudoku puzzles. On the other hand, after training from examples, SATNet could reach near-perfect solving accuracy without the use of ground truth rules. This discrepancy highlights the difficulty in representing and interpreting the underlying logical rules using $S$ alone.

**Limitations of $S$ in representing logical rules.**   We further supplement our empirical observations with theoretical evidence demonstrating the limitations of the clause matrix $S$ in capturing certain logical rules. Let's consider the parity function as an example, which has specific ground truth rules and the corresponding (unnormalized) $S$ matrix defined as follows:

$$\text{Rules in matrix form: } \begin{pmatrix} 1 & 1 & -1 \\ 1 & -1 & 1 \\ -1 & 1 & 1 \\ -1 & -1 & -1 \end{pmatrix}, \quad S = \begin{pmatrix} -1 & 1 & 1 & -1 \\ -1 & 1 & -1 & 1 \\ -1 & -1 & 1 & 1 \\ -1 & -1 & -1 & -1 \end{pmatrix}.$$

In this example, the first two variables in the ground truth rules denote the input bits, while the third variable indicates the parity as the output bit. SATNet, however, introduces an additional truth variable and incorporates it in each clause of $S$ with negative polarity. As a result, the matrix $C = S^T S$ becomes a diagonal matrix $4 \cdot I$, where $I$ is the identity matrix. Given the SDP formulation of SATNet, its optimization remains unaffected by the diagonal elements of $C$, as the calculations associated with these elements result in constants. When all the off-diagonal entries of $C$ are zero, the SDP becomes trivial, thereby preventing SATNet from capturing any meaningful rules from data. This limitation is not unique to the parity function but also extends to other logical rules with similar symmetrical properties. Therefore, the clause matrix $S$ defined in SATNet confronts a challenge in interpretability, as it theoretically fails to represent a range of logical rules.

---

[3]Our detailed experimental settings are provided in Section 4.

## 3.2 Decoding Interpretable Logical Rules from SATNet

Given the lack of interpretability in SATNet's clause matrix $S$, the questions of *why SATNet is capable of learning the underlying logical rules and how we can decode these rules* remain unanswered. To address these questions, we start by revising the optimization formulation of SATNet to a more general form. Building upon this reformulation, we introduce our novel specification approach, "maximum equality", which not only offers a viable interpretation of SATNet but also enables us to derive a set of logical rules in the form of weighted MaxSAT using the learned weights of SATNet.

**Maximum equality specification.** Instead of using the clause matrix $S$, we reformulate Equation 2 in terms of the matrix $C = S^T S$ (with $n_a$ auxiliary variables) and decompose it as:

$$\underset{V \in \mathbb{R}^{k \times (n_d + n_a + 1)}}{\text{minimize}} \sum_{i = \top, j = \top}^{n_d + n_a} c_{ij} \cdot v_i^T v_j, \quad \text{s.t. } \|v_i\| = 1 \text{ for } i \in \{\top, 1, \ldots, n_d + n_a\}. \tag{4}$$

In Equation 4, the objective is to minimize the weighted sum of dot products between all pairs of variable vectors. Considering that all variable vectors are unit vectors, we can view the dot product $v_i^T v_j$ as a measure of similarity between $v_i$ and $v_j$. From this perspective, the objective of Equation 4 is to minimize the similarity between each pair of variable vectors, with the weight $c_{ij}$ dictates the importance of each similarity. Based on this observation, we can further view Equation 4 as a relaxation of the following objective:

$$\underset{Z \in \{0,1\}^{n_d + n_a + 1}}{\text{maximize}} \bigwedge_{i = \top, j = \top}^{n_d + n_a} -2 \cdot c_{ij} \cdot \delta(z_i, z_j). \tag{5}$$

Here, $\delta(\cdot, \cdot)$ is the Kronecker delta function such that $\delta(z_i, z_j) = 1$ when $z_i = z_j$ and 0 else. Equation 5 applies the Kronecker delta function to each pair of Boolean variable assignments $(z_i, z_j)$ and aims to *maximizes* the overall values according to each weight $-2 \cdot c_{ij}$. Note that The factor of 2 is incorporated because the product of two unit vectors $v_i^T v_j$ ranges between $[-1, 1]$, while the Kronecker delta function can only take on values of 0 or 1. We refer to Equation 5 as the "maximum equality" formulation, as it maximizes the weighted equality among pairs of variable assignments.

**Expressive power of maximum equality formulation.** To dive deeper into the maximum equality formulation, we first prove its representational power in expressing any propositional logical rules.

**Proposition 1.** *Our proposed maximum equality formulation can represent arbitrary logical rules within propositional logic, under proper parameterization.*

*Proof.* It is well-established that Max2SAT is NP-Complete [11]. To demonstrate that any propositional logical formula can be reduced to a maximum equality formula, we only need to show the reduction from Max2SAT to maximum equality. Given a clause $(a \vee b)$ in a Max2SAT formula, we can transform it into three maximum equality clauses:

$$-w\,(a = b) \wedge w\,(a = v_\top) \wedge w\,(b = v_\top),$$

where $v_\top$ is the truth variable and $w$ can be any weight value. Similarly, the clauses $(a \vee \neg b)$, $(\neg a \vee b)$, and $(\neg a \vee \neg b)$ can be mapped to three maximum equality clauses, where each satisfying assignment of the original clause has a weight of $2 \cdot w$ more than the unsatisfying one. Compiling these translated clauses gives us a direct transition from a Max2SAT formula to its maximum equality counterpart. While Max2SAT can be transformed to maximum equality by merely introducing a truth variable, general propositional formulas might necessitate more auxiliary variables to reduce to a Max2SAT formula. Therefore, to represent any propositional logical formula using maximum equality, we may also need to incorporate enough auxiliary variables by setting an appropriate value $n_a$. $\qquad \square$

**Transforming maximum equality to weighted MaxSAT.** For a maximum equality formula, we can also transform it into a more standard form, i.e., a weighted MaxSAT formula. Technically, for every clause $w\,(a = b)$ in a maximum equality formula, we convert it into multiple weighted MaxSAT clauses by introducing a new variable $d$ to represent the equality condition $a = b$ as follows:

$$h\,(\neg a \vee b \vee \neg d) \ \wedge \ h\,(a \vee \neg b \vee \neg d) \ \wedge \ w\,(d), \quad \text{if } w > 0,$$
$$h\,(\neg a \vee \neg b \vee d) \ \wedge \ h\,(a \vee b \vee d) \ \wedge \ -w\,(\neg d), \quad \text{if } w < 0,$$

where $h$ denotes the hard constraint, which indicates the infinite weight. Using this transformation, we can decode the learned rules in terms of $C$ to a set of logical rules as a weighted MaxSAT formula.

**Substituting ground truth rules into $C$.** To validate the efficacy of our proposed maximum equality specification as a viable interpretation for SATNet, we directly inject human-written rules into the $C$ matrix, bypassing the use of the clause matrix $S$. This substitution process involves transforming the ground truth rules into a Max2SAT formula and then converting it to the corresponding maximum equality formula. We perform experiments specifically on the parity function, as utilizing the $S$ matrix directly in this case can lead to a trivial SDP formulation for SATNet. As expected, when given the ground truth rules to $C$, SATNet perfectly solves the parity function. This result serves as compelling evidence that the maximum equality formulation could effectively capture and express the logical rules inherent in the problem.

**Learning a sparse $C$ matrix in SATNet.** It is also worth noting that our specification method would result in a weighted MaxSAT formula with approximately $3 \cdot \binom{n+n_a+1}{2}$ clauses if $C$ is a dense matrix that contains no zero elements. For SATNet, $C = S^T S$ is always such a dense matrix. However, solving with such a large number of clauses can be computationally expensive when invoking an exact solver for the weighted MaxSAT formula. To reduce the size of our decoded logical rules, we modify the implementation of SATNet to learn the $C$ matrix directly and further apply the Iterative Hard Thresholding (IHT) algorithm [7, 4] to sparsify it.

Specifically, during each training epoch, we employ a thresholding function with a threshold $\epsilon$ to set some small magnitude elements in $C$ to zero, thereby inducing sparsity. By iteratively applying this thresholding procedure, we progressively obtain a sparser representation of the learned rules, which helps to alleviate the computational burden when solving the weighted MaxSAT problem. The IHT algorithm strikes a balance between the sparsity of $C$ and the performance of SATNet, ensuring that the learned rules remain effective in capturing the underlying logical structure while achieving a more manageable size for the weighted MaxSAT formula. In practice, we can set the threshold $\epsilon$ to be a fraction (e.g., one-fifth) of the average absolute value of non-zero elements in $C$.

## 3.3 Incorporating Domain Knowledge

The interpretability of $C$ makes it feasible to incorporate domain knowledge in many flexible ways. One straightforward approach is to directly add symbolic constraints representing domain knowledge $\phi_{\text{domain}}$ (e.g., a Sudoku puzzle constraint where each cell contains only one digit) to the weighted MaxSAT formula, $\phi_C$, which is derived from the learned matrix $C$. Although the feasibility of introducing domain knowledge in this way is obvious, the reliable interpretability of $C$ plays a central role. Another less obvious but more interesting way to incorporate domain knowledge is to "compile" them into *additive weights*. That is, symbolic knowledge $\phi_{\text{domain}}$ can be properly reduced into weights $\Delta C$, which can be further added to $C$. In fact, it is not hard to show that these two ways are essentially equivalent, i.e., $\phi_{\text{domain}} \wedge \phi_C \equiv \phi_{C+\Delta C}$. In Proposition 1, we have shown that any propositional constraints (e.g., $\phi_{\text{domain}}$) can be reduced into maximum equality constraints (e.g., $\Delta C$), and this reduction guarantees that $\phi_{\text{domain}} \equiv \phi_{\Delta C}$. To prove the equivalence, we only need to show the following property of encoding $C$ into equivalent weighted MaxSAT constraints $\phi_C$.

**Proposition 2.** *For any two matrices $C_1, C_2$ representing maximum equality constraints, we have: $\phi_{C_1} \wedge \phi_{C_2} \equiv \phi_{C_1+C_2}$, where $\phi_C$ represents the weighted MaxSAT constraints reduced from $C$.*

*Proof.* This proposition holds because of the following two properties of weighted MaxSAT constraints: (1) (de-)duplicating hard constraints results in equivalent MaxSAT constraints; (2) weighted soft constraints can be split into multiple constraints (or merged into a single one) as long as the total weight for each unique clause remains unchanged. Considering the matrices $C_1$ and $C_2$ operate as weights of equality relations, which consequently become weights of the underlying MaxSAT constraints, and given that the reduction process ensures that original weights are preserved, we can conclude that the weight matrix $C$ can be split or/and merged without affecting equivalence. □

The interchangeability between symbolic knowledge and the weights of $C$ opens up many interesting interactions between these two different worlds. On one front, domain knowledge can be incorporated into $C$ as the additive weights at various phases, from initialization and training to inference, improving the efficiency and robustness of SATNet. On the other hand, we can also decode the learned matrix $C$ to a set of symbolic constraints, where domain knowledge within propositional logical rules can be directly integrated. In this work, we focus on adding domain knowledge during the inference stage via both approaches.

### 3.4 Verifying Decoded Logical Rules

Once we have the decoded logical rules, the intuitive next step is to formally validate their "correctness" by comparing them with the human-written rules (if available). However, verifying the correctness of a weighted MaxSAT formula based on ground truth expressed in a SAT formula poses inherent difficulties. This complexity stems from two challenges: (1) the lack of a well-defined general equivalence between a SAT formula and a weighted MaxSAT formula, and (2) the potential presence of auxiliary variables in the learned weighted MaxSAT formula that are absent in the ground truth.

To address this issue, we introduce two equivalence definitions between a weighted MaxSAT formula and a SAT formula, based on the concept of *functional equivalence*. At its core, we perceive SAT (or MaxSAT) solving as a function, with a *partial* assignment serving as input, that yields an output, which represents a satisfying (or maximally satisfying) assignment. Informally, to be claimed equivalent, the two formulas should "agree" on all possible input and output pairs. While the intuition behind functional equivalence might seem straightforward, special attention has to be taken into account. This is because SAT (or MaxSAT) solving may not be functional – the output may not be unique or not even exist for a given input. Taking Sudoku as an illustrative example, if the number of input cells is too few, there may exist more than one solution to complete the puzzle; or if the given input cells already have conflicts, there is no way to complete the puzzle.

Here, we consider several formal equivalence definitions as follows.

**Definition 1** (Unique Functional Equivalence). *For any partial assignment to the ground truth expressed in an SAT formula, if a satisfying output exists and is unique, the optimal solution of the weighted MaxSAT formula is also unique and corresponds to the same output (after projecting away auxiliary variables).*

This definition considers functionally dependent input and output (IO) pairs and also suggests a simple enumeration-based method for checking whether the unique functional equivalence holds. We empirically find that such an enumeration-based approach is sufficient for checking small puzzles such as $4 \times 4$ Sudoku, which has around 85,000 functionally dependent IO pairs with minimal inputs.

**Definition 2** (General Functional Equivalence). *For any partial assignments to the ground truth expressed in an SAT formula, if there exists satisfying output(s), the optimal solution(s) of the weighted MaxSAT correspond to a subset of the satisfying assignments of the SAT formula.*

This equivalence definition is stricter than Definition 1 as it additionally constrains the solution space of the weighted MaxSAT formula when the output cannot be uniquely determined by the given input. In this scenario, exhaustively enumerating all possible IO pairs might become prohibitive, even for small puzzles like $4 \times 4$ Sudoku. To address this challenge, we introduce a sufficient condition for proving general functional equivalence, which can be much more efficient to check compared to the enumeration-based approaches.

**Definition 3** (Sufficient Condition for General Functional Equivalence). *Given a weighted MaxSAT formula, if there exists a threshold $\theta$ such that each of its assignments that results in a weight greater (or less) than $\theta$ is meanwhile a satisfiable (or unsatisfiable) solution to an SAT formula, then the general functional equivalence between the two formulas must hold.*

This sufficient condition has a limitation in that it may not always exist for certain weight MaxSAT formulas even if they are functionally equivalent to the ground-truth rules. However, we empirically find that incorporating domain knowledge could often help to make the decoded logical rules meet such a sufficient condition, thereby successfully establishing general functional equivalence.

## 4 Experiments

We present the evaluation results in this section. We test our proposed framework on the tasks of stream transformations and Sudoku puzzles. We train both original SATNet [32] and our modified version SATNet* to learn the underlying logical rules in the data and decode them into a weighted MaxSAT formula using our specification approach. Given the generated rules, we utilize Gurobi [13] for exact inference and use the state-of-the-art MaxSAT solver CASHWMaxSAT [16] for verification. All experiments are carried out on a single RTX8000 GPU and 16 CPU cores across 100 epochs, using the Adam optimizer with a learning rate of $2 \times 10^{-3}$. All other hyperparameters are also set the same as in the paper [32].

## 4.1 Stream Transformations

**Experiment setup.** In this experiment, we evaluate our proposed framework on three stream transformation tasks, i.e., the parity function, binary bit addition, and symbol counting. The parity function is the chained XOR of an input 0-1 stream. Following SATNet [32], we set the length of the input stream to 20 and 40 and generate a dataset of 10,000 examples for each length. The binary bit addition task aims to calculate the sum of two binary strings with the corresponding carry bit. We set the input length to 10, 20, and 30 and randomly sample 4,000 instances for each length. The symbol counting adds the occurrence number $N$ of one specific symbol (e.g., bit 1) of one stream and computes $N$ modulo $k$. The input stream has a length of 40, and we explore different values of $k$, namely 2, 4, 8, and 16. For each value of $k$, we randomly generate a dataset comprising 1,000 samples. For all these datasets, we split them into training/testing sets with 90%/10% proportions.

**Results.** On all three datasets, both SATNet and SATNet* can learn the underlying logical rules in the data perfectly, solving all testing instances. Since valid (partial) input spaces for these tasks are relatively small and can be efficiently enumerated, we verify the decoded rules with the ground truth using the *general functional equivalence*. As expected, given any partial input, the optimal solutions of our decoded weighted MaxSAT formula align with the satisfying assignments in the ground truth rules. This result shows that our decoded rules from the matrix $C$ are verifiably reliable.

## 4.2 Sudoku Puzzles

**Experiment Setup.** In this experiment, we evaluate the performance of our approach on Sudoku puzzles of varying board sizes: $4 \times 4$ and $9 \times 9$. For the $4 \times 4$ Sudoku puzzles, we construct a dataset by enumerating all possible puzzles that have a unique solution with minimal inputs, resulting in a total of 85,632 puzzles. To ensure proper evaluation, we split this dataset into training and testing sets with a ratio of 9:1. For the $9 \times 9$ Sudoku puzzles, we use the same dataset in SATNet [32], consisting of 9,000 instances for training and 1,000 instances for testing. It is also noteworthy that the original experiment in SATNet implicitly enforces that each group of variables corresponding to the same cell has exactly one value of 1 during inference. However, our experiments discard these constraints and instead round all variable values based on a threshold for both SATNet and SATNet*.

**Results.** On the $4 \times 4$ Sudoku dataset, both SATNet and SATNet* exhibit remarkable performance, achieving testing accuracies of 99.89% and 99.90%. Notably, the learned $C$ matrix in SATNet* is much sparser than that in SATNet, containing approximately 53% zero elements. On the other hand, when leveraging Gurobi to solve the weighted MaxSAT formulas derived from SATNet and SATNet*, we can achieve perfect accuracy on the testing set. To further validate the robustness of our decoded rules, we consider the *unique functional equivalence* between our decoded MaxSAT formula and the human-written rules and successfully verify such a functional equivalence. This result demonstrates that our decoded logical rules are highly reliable, despite originating from SATNet and SATNet*, which inherently involves some errors during the approximate solving process.

Furthermore, we conduct experiments to add domain knowledge into our framework during inference. Specifically, by incorporating the *partial* Sudoku constraints (one cell cannot contain more than one digit) to our decoded weighted MaxSAT formula, we are able to efficiently verify that our augmented logical rules satisfy the *sufficient condition for general functional equivalence* with the ground truth rules. Alternatively, if we integrate this knowledge into the weight matrix $C$ as a "weight repair", SATNet* can also successfully solve all Sudoku puzzles in the testing set, rectifying the previously failed cases. These results demonstrate the flexibility and adaptability of our approach when incorporating additional symbolic knowledge.

On the $9 \times 9$ Sudoku dataset, SATNet achieves a testing accuracy of 88.30% and decodes into a weighted MaxSAT formula with hundreds of thousands of clauses [4]. Given the enormous size of the decoded rules, both Gurobi and CASHWMaxSAT fail to verify or find a counterexample in hours. Therefore, in this case, we only compare the weight of the failed predictions against the ground truth one (with the same assignments of auxiliary variables). Our results show that the wrong predictions always have strictly lower weights than the ground truth ones, demonstrating that the mistakes of SATNet are due to the suboptimal solutions according to our decoded weighted MaxSAT formula.

---

[4]We also experiment with SATNet* but find its convergence rate is slower than that of SATNet in 100 epochs and the resulted number of clauses is still in the hundreds of thousands. As such, we use SATNet in this scenario.

# 5    Related Work

**Differentiable learning of logical rules.**    Our work is closely related to recent research that focuses on learning underlying logical rules from data in an end-to-end manner, without relying on pre-specified rule templates. Several approaches [1, 32, 6, 21, 28, 3] formalize the task of learning logical rules as a constrained optimization procedure, using different relaxations on logical operators to solve it. For example, OptNet [1] models the problem as a quadratic program and learns $4 \times 4$ Sudoku rules from input-output games, while SATNet [32] and its variant [18] apply differentiable SDP relaxations to solve $9 \times 9$ Sudoku puzzles and Rubik's cube. Some alternative works [2, 27, 9, 37] view the rule learning process as a black box and utilize neural networks as function approximators to learn logical rules. These efforts include Neural Logic Machines [9], which uses simple multi-layer perceptrons to recover lifted rules in first-order logic, and Recurrent Transformers [37], which employs an extended version of transformers to solve both textual and visual Sudoku puzzles [5, 29]. However, all these attempts embed the learned rules into weights of neural networks, failing to give human-readable interpretations and making them infeasible for verification. To our best knowledge, our proposed framework is the first attempt to decode a set of interpretable logic structures through differentiable learning, and formally verify their functional correctness.

**Differentiable logic programming.**    Logic programming [19] is a programming paradigm that expresses facts and rules about problems within a system of formal logic. Recently, there have been tremendous efforts to combine neural networks with logic programming, enabling neural networks to generate explicit logical programs from examples. A line of research [25, 34, 10, 14, 26, 35] augments inductive logic programming (ILP) [22] with neural networks. For instance, Neural LP [34] learns first-order logical rules for knowledge base reasoning, while $\partial$ILP [10] generalizes ILP to learn robust rules from noisy data. Besides ILP, some approaches integrate neural networks with other types of logic programming: DeepProbLog [20] extends Problog [24] by learning the weights for each probabilistic logical rule, ABL [8] and NeuroLog [30] entangle abductive logic programming [15] with machine learning models through gradient-descent based optimization, and NeurASP [36] applies a similar idea to answer set programming [17]. While the integration of logic programming with neural networks has shown promise in deriving interpretable logical rules, a common challenge in these approaches is the reliance on predefined rule templates. This requirement often necessitates significant expertise and careful design, making it a highly non-trivial task. In contrast, our proposed framework eliminates the need for such templates, offering a more flexible and efficient approach to learning interpretable logical rules from data.

# 6    Discussions

**Limitations and future work.**    There are a few limitations to our proposed framework. Firstly, our current framework builds on SATNet, which is limited to learning propositional logical rules and cannot handle rules in first-order or higher-order logic. Additionally, if SATNet fails to capture the underlying logic structures in the data, our decoded rules may be incorrect and unreliable. Secondly, given the number of variables $n$, our approach may result in a weighted MaxSAT formula with $\mathcal{O}(n^2)$ clauses in the worst case. This can potentially lead to time-consuming inference processes, especially when utilizing an exact solver for large values of $n$ in practical scenarios. We would like to expand our approach to support more expressive rule representations and explore alternative optimization approaches to reduce the size of our learned logical rules as future work.

**Conclusion.**    In this work, we present a novel framework for extracting interpretable logical rules from the learned weights in SATNet. We demonstrate, both empirically and theoretically, that the original formulation in SATNet lacks clear interpretability. To address this, we introduce maximum equality specification, which enables the interchangeability between the weights of SATNet and a set of logical rules in the weighted MaxSAT form, as well as the flexible integration with domain knowledge. To verify the correctness of our decoded formula, we introduce multiple verification approaches that compare the solution space of our generated rules with the ground truth. Experimental evaluations conducted on the stream transformations and Sudoku puzzles demonstrate that the rules generated by our framework are highly accurate and formally verifiable. We hope our approach will lay the groundwork for bridging deep learning and interpretable rule learning and stimulate further innovation in this exciting field.

## Acknowledgments and Disclosure of Funding

We thank the anonymous reviewers for their insightful comments. This work was supported, in part, by Individual Discovery Grants from the Natural Sciences and Engineering Research Council of Canada, and the Canada CIFAR AI Chair Program.

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
