# OpenReview forum: "Learning Reliable Logical Rules with SATNet"
_NeurIPS.cc/2023/Conference — NeurIPS 2023 poster_

### Official Review · Reviewer_fiQw · 2023-06-10

**Soundness:** 3 good
**Presentation:** 3 good
**Contribution:** 3 good
**Rating:** 7
**Confidence:** 3

**Summary:**

The authors propose a new framework to generate interpretable and verifiable logical rules through differentiable learning. The framework is built upon SATNet, but the paper proposes a new interpretation method called maximum equality to decode the weights of SATNet into logical rules. The paper also proposes several verification methods to validate the decoded rules against ground truth rules. Experiments show that the decoded rules are highly reliable and functionally equivalent to the ground truth rules.

**Strengths:**

- The impressive contribution of this paper is to distill the black box weights of a differentiable logic program like SATNet into human interpretable rules (boolean expressions) that can be verified.
- Moreover, while the initial SATNet itself fails to achieve 100% test accuracy, the decoded interpretable rules manage to achieve perfect 100% test accuracy. This is remarkable because in the realm of logical reasoning, true generalization is a binary outcome: either the logical rules have truly been learned or they have not. In this paper, they show that they indeed have been learned, and verifiably so.

**Weaknesses:**

- A major caveat to this work is that the 100% test accuracy was achieved by adding symbolic constraints that represent domain knowledge (e.g. rules of Sudoku). However, this is not necessarily a weakness, because incorporating domain knowledge is a useful thing to do in general, and it's great that the authors propose a recipe for doing so.
- It would be nice to see the authors apply their technique to visual Sudoku. That was one of the hallmark contributions of the original SATNet paper that learned to play Sudoku by looking at images of digits.

**Questions:**

- The paper mentions that the thresholding function was set to be approximately one-fifth of the average absolute value of non-zero elements in C. How was this decided, and how sensitive is the performance relative to the chosen threshold? It would be nice to see a graph of this.

**Limitations:**

The limitations section was well-written. If possible, I'd like to see an extended discussion of future work (perhaps in the Appendix), so the authors can provide useful directions for future research.

---

> ### Author Rebuttal · Authors · 2023-08-07
>
> Thank you for the positive feedback and comments! In the following, please let us address your raised concerns and questions.
>
> > The 100% test accuracy was achieved by adding symbolic constraints that represent domain knowledge (e.g. rules of Sudoku).
>
> We want to clarify that our approach could achieve 100% accuracy *without* adding any additional constraints on the 4 $\times$ 4 Sudoku dataset when employing the exact solver Gurobi on our decoded logical rules, and we can also verify the decoded rules satisfy the *unique functional equivalence* with the ground truth rules. Given the vast input space of 4 $\times$ 4 Sudoku puzzles, exhaustively enumerating all IO pairs for *general functional equivalence* verification is infeasible. Instead, we discover that by incorporating *partial* Sudoku rules (e.g., the constraints that each cell can only contain one digit) into our decoded rule sets, we can efficiently verify that our logical rules satisfy *the sufficient condition* for general functional equivalence. Furthermore, in practical real-world scenarios, we acknowledge the potential benefit of utilizing *partial* domain knowledge/commonsense knowledge (as prior symbolic constraints). By applying our approach to learn interpretable rules from data and subsequently adding this domain knowledge into our rule sets, we can further enhance the reliability of our decoded logical rules. This adaptability to integrate domain-specific information makes our approach more versatile and suitable for real-world applications.
>
> > It would be nice to see the authors apply their technique to visual Sudoku.
>
> Thanks for the suggestions. We conduct additional experiments on 4 $\times$ 4 visual Sudoku datasets. Similar to the non-visual setting, using SATNet and SATNet* alone could achieve a solving accuracy of 99.83\% and 99.28\% respectively, whereas using exact solver Gurobi on our decoded rules could achieve 100\% accuracy. Moreover, we further verify that the decoded rules in this setting satisfy the *unique functional equivalence* as well.
>
> > How was the threshold function for sparsity decided, and how sensitive is the performance relative to the chosen threshold?
>
> The threshold function we design is based on our empirical evaluation. During our experiments, we observe that the expressivity of the decoded rules remains relatively stable across different threshold values. To validate this further, we conduct additional experiments using various threshold values for SATNet* on the 4 × 4 Sudoku dataset. The results are summarized below (denoting the average nonzero elements of $C$ as $\mu$):
>
> | threshold | Sparsity | Solving Accuracy with Gurobi | Solving Time with Gurobi (per instance)|
> | :-: | :-: | :-: | :-: |
> | 0.05 $\mu$ | 0.12 | 100\% | 4.37s |
> | 0.10 $\mu$ | 0.15 | 100\% | 3.91s |
> | 0.20 $\mu$ | 0.53 | 100\% | 0.92s |
> | 0.25 $\mu$ | 0.53 | 100\% | 0.93s |
> | 0.50 $\mu$ | 0.57 | 100\% | 0.66s |
> | 0.60 $\mu$ | 0.74 | 100\% | 0.37s |
>
> > More discussion on future work.
>
> We further discuss one possible way to improve our work as future work. As stated in our paper, our approach may result in a weighted MaxSAT formula with $O(n^2)$ clauses in the worst case. One promising avenue for improving our approach is to address the challenge of learning lifted rules. By effectively learning and incorporating lifted rules into the model, we can significantly reduce the size of the resulting rule sets. For example, representing concepts such as rows, columns, and $3 \times 3$ squares on the Sudoku board as variables would allow us to capture more concise and interpretable rules. Accomplishing this involves developing a domain-specific language (DSL) that efficiently represents a wide range of logical rules. The designed DSL would enable the lifting process, allowing SATNet to learn and leverage high-level concepts and relationships, leading to improved performance and compact decoded rule sets.

---

> > ### Comment · Reviewer_fiQw · 2023-08-13
> >
> > Thanks for the response.

---

### Official Review · Reviewer_sq5Z · 2023-06-19

**Soundness:** 4 excellent
**Presentation:** 4 excellent
**Contribution:** 4 excellent
**Rating:** 8
**Confidence:** 3

**Summary:**

This paper builds on the SATNet framework on MaxSAT problems, adds an interpretation method that allows the conversion between its weight and propositional logical rules. Effective verification methods are proposed to see if the decoded rules from SATNet are functionally equivalent to the ground truth. Experiments on stream transformation and Sudoku problems show that the decoded "hard" logical rules achieve better performance than SATNet itself. The logical rules are verified to be equivalent to the ground truth.

**Strengths:**

- The proposed interpretation method is well motivated and grounded theoretically.
- The proposed interpretation method allows for the integration of symbolic knowledge as additive weights, which is flexible for the injection of human domain knowledge.
- The verification methods are sound and have been practised on the experimental tasks of stream transformations and Sudoku.
- The paper is well-written.
- The decoded logical rules outperform SATNet on the experimental tasks, showing its practical effectiveness.

Overall, it's a solid contribution to the interpretability of deep learning and neuro-symbolic computation communities.

**Weaknesses:**

For the machine learning readers, perhaps the authors can give a bit more elaboration when propositions are stated. A couple of worked examples can also help the reader understand a lot better.

**Questions:**

What challenges do you foresee when extending this work to more expressive logics like FOL and HOL, apart from the scaling issues mentioned in the discussions section? Are there fundamental bottlenecks to this? I would like to see more discussions on this, as propositional logic seems slightly limiting.

**Limitations:**

The authors addressed the limitations and scoped the work well.

---

> ### Author Rebuttal · Authors · 2023-08-07
>
> Thanks a lot for your positive review and valuable feedback! We would elaborate on our propositions in our revision. In this response, we discuss the challenges we anticipate when extending our work to more expressive logic like FOL and HOL:
>
> * Representing logic rules in FOL/HOL using a simple matrix form and formulating the entire procedure as a clear mathematical optimization becomes challenging. Unlike propositional rules, these logics involve more complex structures (especially with quantifiers) that are difficult to encapsulate within a continuous optimization framework. To extend our work to more expressive logic, we anticipate the need for innovative approaches that incorporate efficient *quantifier elimination* as well as *lifted inference* techniques into the differentiable optimization process.
>
> * FOL/HOL entails a significantly larger rule space compared to propositional logic, making it difficult to learn from data *without predefined constraints*. The expanded search space poses challenges in identifying meaningful and interpretable rules from the vast possibilities. Addressing this challenge may involve developing a domain-specific language (DSL) / meta rules to limit the rule space.
>
> * Verifying the learned rules against ground truth rules in FOL/HOL is not straightforward. Due to the inherent complexity of these logics, the design of the verification process requires more careful consideration. Proving the equivalence of two FOL/HOL logical rules may necessitate the use of theorem provers (e.g., Coq), which requires considerable expertise. Beyond scalability, automating the verification process presents another challenge.
>
> For example, one potential future work is to explore building specification/interpretation methods on the top of Neural Logic Machines (NLM) [1], which focuses on learning FOL rules with only nullary, unary, and binary predicates from data. However, unlike SATNet, which stores learned rules in a single $S$ (or $C$) matrix, NLM stores rules in the weights of neural networks (several MLPs). The optimization objective in NLM is not as straightforward as SATNet's, which can be expressed as a SDP. Additionally, interpreting MLPs is more challenging than interpreting a matrix alone. To decode interpretable rules from NLM, one may explore replacing MLPs with learnable matrices and transforming the entire neural architecture into a clear mathematical optimization form. By doing so, our similar specification methods may be applied to the revised NLM to derive interpretable rules effectively.
>
> [1] Dong, Honghua, et al. "Neural Logic Machines." In International Conference on Learning Representations (ICLR), 2019

---

> > ### Comment · Reviewer_sq5Z · 2023-08-17
> >
> > Thank you for the response. I will maintain my score.

---

### Official Review · Reviewer_cdcd · 2023-06-29

**Soundness:** 2 fair
**Presentation:** 4 excellent
**Contribution:** 3 good
**Rating:** 6
**Confidence:** 4

**Summary:**

The authors show that interpreting the SATNet model is not reliable. They reformulate the SATNet objective to a 'maximum equality specification' on matrix $C$. Assuming $C$ is ternary instead of continuous, the objective can be interpreted as a MaxSat problem. This problem can then be verified. Furthermore, the new objective is amenable to injecting symbolic background knowledge.

Post rebuttal: I appreciate in-depth responses of the authors. I will keep my score.

**Strengths:**

The paper is well-written and easy to follow. The structure is excellent, interleaving some experiments to motivate some choices. The problem of interpretability of learned rules is important and often overlooked in these logic-inspired architectures. I do not believe SATNet has been studied in this context before.
The paper presents interesting failure cases of SATNet. The symbolic background knowledge injection through compiling inside matrix $C$ is a very useful result, increasing the model's applicability.

**Weaknesses:**

I have some questions about the theoretical ideas presented in the paper, particularly on the limitations of SATNet. The experiments are somewhat simple and could be presented more clearly or on more tasks (for instance, on ILP benchmarks like in $\partial$ILP [1]).

- The experiments could be a lot clearer with a table for the different tasks and the different evaluations. Ie, Compare SATNet and SATNet* as a continuous model and in the MaxSAT version.
- The benefits of the maximum equality specification for interpretability are not entirely apparent to me (see questions).

[1] Evans, R., & Grefenstette, E. (2018). Learning explanatory rules from noisy data. Journal of Artificial Intelligence Research, 61, 1-64.

**Questions:**

- Limitations of $S$ (p4, 145-155): The matrix given does not represent parity / XOR. I think coordinate (3, 4) should be a -1 (ie the last row is -1 -1 -1). Otherwise the (correct) input $110$ is rejected. On this corrected matrix $S$, I computed $S^T S$ and did not get a zero matrix as claimed but a diagonal matrix ($4 I$ in particular). This is not due to the correction, as $S_\top^T S_\top = 4$, so coordinate (1, 1) is necessarily 4.
- Same part: The claim is that since $S^T S$ collapses, SATNet cannot theoretically represent a range of logical rules. I think this claim should be made a bit stronger: What (I think) you want to show is that two non-semantically equivalent CNF formulas map to the same $S^T S$. For instance, is there another problem than parity for which $S^T S$ equals $4 I$?
- Have you tried interpreting the matrix $S$ instead when using the sparsification procedure? Instead of the MaxSAT procedure? Would that be worse than the MaxSAT interpretation?
- Experiments: What exactly makes SATNet* different from SATNet? The sparsification?
- 348: Why is SATNet* not evaluated on 9x9 sudokus?

**Limitations:**

While the authors developed a sparsification method, the MaxSAT problem will likely be huge and hard to interpret or verify. The verification part of this limitation is mentioned, not the interpretability.

The verification method amounts to enumerating IO Pairs. I would have hoped the interpretable rules would allow symbolic reasoning to prove equivalence. The authors argue this is impossible because it would require a transformation of SAT into MaxSAT (or vice-versa).

---

> ### Author Rebuttal · Authors · 2023-08-07
>
> Thank you for the detailed and in-depth comments and questions! In the following, we hope to address the stated weaknesses and questions of our paper.
>
> > The experiments are somewhat simple and could be presented more clearly or on more tasks like ILP benchmarks in $\partial$ILP.
>
> We want to clarify that the setting of SATNet is quite different than that in ILP: SATNet aims to learn propositional logical rules directly from data without any predefined rules, while ILP requires background knowledge and predefined rules (or templates). Thus, applying SATNet to ILP tasks might not be fair for comparison due to the contrasting problem settings.
>
> > Limitations of $S$ / The claim that SATNet cannot theoretically represent a range of logical rules.
>
> We apologize for the typos in our paper and appreciate your attention to detail. The last row in the ground truth rules should be (-1, -1, -1), and the corresponding matrix $C = S^TS = 4 I$. However, what we are particularly concerned about are the off-diagonal elements in the matrix $C$. Due to the SDP formulation of SATNet, its optimization is not affected by the diagonal elements of $C$, as the calculations associated with these diagonal elements lead to constants (specifically, $v_i^Tv_i$ = 1). When all the off-diagonal elements of $S^TS$ are zero, the SDP becomes trivial, and SATNet fails to capture any meaningful logical rules. This issue also exhibits in other symmetric rules like $(\neg x \lor y) \land (x \lor \neg y)$. In our revision, we will provide a more detailed explanation of this aspect to enhance clarity.
>
> > Have you tried interpreting the matrix instead when using the sparsification procedure?
>
> We initially attempted several regularization methods to enforce $S$ to be a ternary matrix during the training process, but unfortunately, these approaches did not yield successful results.
>
> > What exactly makes SATNet* different from SATNet? The sparsification?
>
> We want to first emphasize that SATNet* and SATNet can be both used to learn rules from data and our specification methods (with exact inference) and verification methods can be applied to both of them. Here are two main technical differences between SATNet* and SATNet:
>
> * Learning procedure: SATNet learns the $S$ matrix, while SATNet* considers the $C$ matrix as the parameters and directly learns the weights of $C$. This distinction leads to a reduction in both the number of parameters and hyperparameters in our model. Specifically, $S \in \mathbb{R}^{m \times n}$ and $C \in \mathbb{R}^{n \times n}$, where $n$ is the number of variables (including the auxiliary variables), and $m$ is the number of clauses. In many cases, $m > n$, which means that SATNet* may have fewer parameters than SATNet. Moreover, when setting hyperparameters, SATNet considers the number of clauses and auxiliary variables, while SATNet* only considers the number of auxiliary variables. This makes SATNet* a more convenient and practical choice for real-world applications.
>
> * The Sparsification: The sparsification technique is another key distinction. It strikes a balance between the expressivity and the size of the learned logical rules. By employing sparsification, SATNet* can reduce the size of decoded logical rules in some cases, which in turn accelerates exact inference using Gurobi (with more than 10$\times$ speed up).
>
> > Why is SATNet* not evaluated on 9 $\times$ 9 sudokus?
>
> We did attempt to use SATNet* on 9 $\times$ 9 sudokus; however, the convergence rate of SATNet* was slower than SATNet in 100 epochs. Although SATNet* could indeed reduce the size of decoded logical rules, the number of clauses is still in the hundreds of thousands. As a result, we chose to use SATNet instead for $9 \times 9$ Sudokus. Besides 9 $\times$ 9 Sudoku datasets, we find that SATNet* could converge as quickly as SATNet and could effectively reduce the size of decoded rules.

---

> > ### Comment · Reviewer_cdcd · 2023-08-14
> >
> > I thank the authors for their comment.
> >
> > I am not entirely sure I agree with the assessment of ILP there. ILP does not _require_ background knowledge, it is optional in most cases. From what I recall, SATNet also uses 'templates', in that it creates propositional CNFs. The problem settings thus seem pretty similar to me: Both aim to learn interpretable rules.
> >
> > Thanks for the clarification on the example, and I am somewhat glad I did not make a calculation mistake!
> >
> > About 9x9 sudokus: I would appreciate if the mentioned results on slow convergence are added for completeness to the paper. Also, why is the number of clauses in the 100s of thousands? That seems like more than I would have expected. Do you believe the slowness of SATNet* to come because it is not 'overparameterised', in a sense?

---

> > > ### Author Response · Authors · 2023-08-14
> > >
> > > Thank you for your quick and insightful feedback! We hope to address your comments in detail as follows:
> > >
> > > > ILP does not require background knowledge, it is optional in most cases. From what I recall, SATNet also uses 'templates', in that it creates propositional CNFs.
> > >
> > > While it is true that background knowledge is not always mandatory for ILP, tasks where ILP is employed typically integrate this knowledge in practice (e.g., on the datasets in $\partial$ ILP). Besides this, the main difference between ILP and SATNet is their rule structures. In ILP, human experts need to meticulously define rule templates & program templates *on the predicates* to constrain the rule space. Conversely, SATNet doesn’t demand such intricate templates for the underlying logic structures, focusing instead on learning the propositional logical rules *associated with the variables*. For SATNet, the only requirement is to set the shape of the $S$ matrix, particularly determining the number of variables, $n$ + $n_a$, and the number of clauses, $m$. Therefore, the settings and applications of ILP and SATNet are quite different.
> > >
> > > > About 9x9 sudokus: I would appreciate it if the mentioned results on slow convergence are added for completeness to the paper.
> > >
> > > We will incorporate these mentioned results into the revision of our paper.
> > >
> > > > Why is the number of clauses in the 100s of thousands? That seems like more than I would have expected.
> > >
> > > As stated in our paper, if $C$ is a dense matrix without zero elements, our specification method would result in a weighted MaxSAT formula with around $3 \cdot \binom{n+n_a+1}{2}$ clauses. In the case of $9 \times 9$ Sudoku, with a defined variable $n = 729$ and an auxiliary variable $n_a = 300$, the decoded weighted MaxSAT formula would yield such a massive number of clauses.
> > >
> > > > Do you believe the slowness of SATNet* to come because it is not 'overparameterized', in a sense?
> > >
> > > Yes, we believe that your raised point aligns with our thoughts and might be a reason regarding the performance of SATNet*. As stated in our paper, the sparsification procedure strikes a balance between the expressivity of SATNet* to learn the underlying rules and the size of the decoded rules. An interesting perspective of the results might be that SATNet, being 'overparameterized', effectively captures embedded rules within its parameters during training, whereas SATNet* falls short on $9 \times 9$ Sudoku puzzles.
> > >
> > > We genuinely appreciate your feedback and are open to further discussions to enhance our work!

---

> > > > ### Comment · Reviewer_cdcd · 2023-08-19
> > > >
> > > > > In ILP, human experts need to meticulously define rule templates & program templates on the predicates to constrain the rule space. Conversely, SATNet doesn’t demand such intricate templates for the underlying logic structures, focusing instead on learning the propositional logical rules associated with the variables.
> > > >
> > > > Can I understand this as saying that SATNet acts on a propositional formalism while ILP acts on logic programming, which is much stronger than propositional? ILP tends to generalise better because it has to 'compress' its description.
> > > >
> > > > > We will incorporate these mentioned results into the revision of our paper.
> > > >
> > > > Happy to hear that!
> > > >
> > > > > As stated in our paper, the sparsification procedure strikes a balance between the expressivity of SATNet* to learn the underlying rules and the size of the decoded rules. An interesting perspective of the results might be that SATNet, being 'overparameterized', effectively captures embedded rules within its parameters during training, whereas SATNet* falls short on 9x9 Sudoku puzzles.
> > > >
> > > > Interesting, maybe so! That may be worth discussing in the results.
> > > >
> > > > I appreciate in depth responses of the authors. I will keep my score.

---

> > > > > ### Author Response · Authors · 2023-08-21
> > > > >
> > > > > > Can I understand this as saying that SATNet acts on a propositional formalism while ILP acts on logic programming, which is much stronger than propositional? ILP tends to generalize better because it has to 'compress' its description.
> > > > >
> > > > > Yes, the original SATNet can be viewed as operating within a propositional formalism, focusing on learning logical rules in a 'low-level' and 'fine-grained' fashion. ILP, on the other hand, aims to generate a logic program on some 'high-level' concepts and relationships (i.e., predefined predicates), which can learn much more 'compact' logical rules. One potential future work is to design a domain-specific language (DSL) for SATNet* that enables it to learn and leverage high-level concepts and relationships, leading to more concise decoded rule sets.
> > > > >
> > > > > We are truly grateful for the reviewer's insightful feedback and engaging discussion!

---

### Official Review · Reviewer_un3V · 2023-07-07

**Soundness:** 3 good
**Presentation:** 3 good
**Contribution:** 3 good
**Rating:** 7
**Confidence:** 2

**Summary:**

This paper builds on SATNet, a differentiable MaxSMT solver that was proposed in the past to learn logical rules from input-output examples. SANet is based on a "low rank semidefinite programming approach" and uses a learnable matrix S to capture the logical rules. SATNet was shown to learn to solve e.g. Sudoku puzzles with near perfect accuracy. The problem addressed in this paper is that the S matrix and the learned weights in SATNet lack a clear, logical meaning. The authors first describe a set of (failed) experiments that aim to extract the meaning from S. Then they describe a new formulation called maximum equality which enables them to formulate MaxSMT formulas that can be solved with off-the-shelf solvers. The authors demonstrate both theoretically and empirically that this new formulation indeed captures the underlying logical rules; it also opens up the possibility of new applications, such as adding domain specific constraints to a problem

**Strengths:**

* The paper is very well written. It provides a gentle introduction to teh problem and a clear running example to illustrate the technique.
* The paper makes a clear contribution over the previous SATNet approach.The proposed approach is very interesting and has many practical applications.
* The novelty is not only in the "decoding" but also in teh functional comparison between the decoded rules and teh ground truth.

**Weaknesses:**

none

**Questions:**

none

**Limitations:**

The paper describes carefully the limitations of the proposed approach.

---

> ### Author Rebuttal · Authors · 2023-08-07
>
> Thanks a lot for your positive comments and feedback! If you have further questions, we are happy to answer them!

---

### Official Review · Reviewer_ASVp · 2023-07-29

**Soundness:** 3 good
**Presentation:** 2 fair
**Contribution:** 2 fair
**Rating:** 4
**Confidence:** 3

**Summary:**

The work investigates the problem of generating interpretable and verifiable logical rules through differential learning. Specifically,  a deep network layer for satisfiability solving (SATNet) in a differentiable maximum satisfiability (MAXSAT) solver that learns logical rules from input-output examples was used. Through experiments, the authors claim that the learned weights in SATNet lack explainability. To address this issue, a method called maximum equality is proposed to interpretate the weights of SATNet into a set of propositional logical rules. Stream transformations and Sudoku problems were taken as the tasks to evaluate the method.

Post rebuttal: I have read the authors' rebuttal and I appreciate the authors' effort in addressing my concerns. I agree with the authors' point on interpretable logical rules in the rebuttal. My concerns on the application of the proposed methods to models beyond SATNet and other more complex tasks still hold.

**Strengths:**

+The work proposes a method called maximum equality to interpretate the weights of SATNet into a set of propositional logical rules to improve the explainability of SATNet.
+The method was evaluated on the Stream transformations and Sudoku tasks and the results show that decoded rules from the maximum equality method are better than the ones out of the original SATNet.
+To a certain extent, it improves the explainability of learned logical rules from input-output examples.

**Weaknesses:**

-The work builds upon a specific satisfiability solving method (SATNet), which limits the impact of the work.
-It is claimed that the lack of interpretation of SATNet was due to that the clause matrixS is not enforced as ternary and a method of rounding S to a ternary matrix didn't yield correct logical rules. However, it would be interesting to investigate how to interpretate S through a learnable network or another approach.
-The matrix C = S^T S is decomposed, and the objective of maximum equality is to minimize the similarity between each pair of variable vectors of S. A weight c_{ij} is associated with the similarity between each pair of variable vectors. It is not clear how c_{ij} will affect the generated logical rules.
-The work relies on the prior work SATNet and a few terms are not explained, which hinders understanding the work. For example, SATNet -   satisfiability solving network; MaxSAT - maximum satisfiability (MAXSAT) solver.
-In Eq. 2 and Eq. 3, mimimize -> minimize

**Questions:**

-Would it be feasible to use a deep neural network to interpret S of SATNet?
-Is c_{ij} predefined or learnable?
-How is an appropriate value set for n_a?
-The visual Sudoku problem was used in the SATNet work, have you tested your maximum equality method on this task as well?

**Limitations:**

The authors have mentioned the limitation of this work in only learning propositional logical rules and not being able to handle rules in first-order or higher-order logic.

---

> ### Author Rebuttal · Authors · 2023-08-07
>
> Thanks for your comments and feedback. In the following, we will address your questions point by point.
>
> > The work builds upon a specific satisfiability solving method (SATNet), which limits the impact of the work.
>
> Indeed, SATNet is an award-winning architecture (ICML 2019 Best paper Honorable mention) that achieves state-of-the-art performance for learning *neural representations* of logical rules from data without using specified rule templates. While our approach builds upon SATNet, it is important to highlight that our work represents the *first* contribution in the logical rule learning/logic programming field, enabling the learning of *interpretable* and *verifiable* logical rules directly from data without any prior rule templates. By focusing on interpretability, our work expands the potential applications of differential logical rule learning that requires the transparent solving process and provides a novel perspective in this domain.
>
> > It would be interesting to investigate how to interpret $S$ through a learnable network or another approach / Would it be feasible to use a deep neural network to interpret $S$ of SATNet?
>
> You raise an interesting point about interpreting the $S$ matrix through a learnable network. However, it is essential to clarify that interpretable logical rules aim to be human-readable, enabling white-box problem-solving processes. While neural networks themselves are challenging to interpret, decoding reliable human-readable rules directly from the $S$ matrix using neural networks becomes even more complex. To our best knowledge, there is no existing learnable approach that can reliably interpret $S$ to generate a set of logical rules in an interpretable form. Our approach naturally aligns with the optimization process of SATNet and enables the *interchangeability* between weights and propositional logical rules, providing a viable and effective solution for decoding a set of interpretable rules.
>
> > It is not clear how $c_{ij}$ will affect the generated logical rules / Is $c_{ij}$ predefined or learnable?
>
> In SATNet*, the only learnable parameter is the $C$ matrix, which is fully learned from the data. Each element $c_{ij}$ contributes to the weights in our maximum equality formulation.
>
> > The work relies on the prior work SATNet and a few terms are not explained, which hinders understanding the work.
>
> Thanks for pointing out this issue and our typos. In the revised version of our paper, we will provide a comprehensive explanation of SATNet and clarify all relevant terms to ensure a better understanding for readers.
>
> > How is an appropriate value set for n_a?
>
> The value of $n_a$ is set differently for various problems. Generally, $n_a$ can be chosen around the number of defined variables ($n$). In our experiments, we set $n_a$ to the same value as the original SATNet paper to ensure a fair comparison.
>
> > The visual Sudoku problem was used in the SATNet work, have you tested your maximum equality method on this task as well?
>
> Thanks for the suggestions. We conduct additional experiments on 4 $\times$ 4 visual Sudoku datasets. Similar to the non-visual setting, using SATNet and SATNet* alone could achieve a solving accuracy of 99.83\% and 99.28\% respectively, whereas using exact solver Gurobi on our decoded rules could achieve 100\% accuracy. Moreover, we further verify that the decoded rules in this setting satisfy the *unique functional equivalence* as well.

---

> > ### Comment · Reviewer_ASVp · 2023-08-15
> > **Rebuttal acknowledged**
> >
> > I thank the authors for their replies.

---

> > > ### Author Response · Authors · 2023-08-18
> > >
> > > Thank you for the acknowledgment of reading our responses. We appreciate that the questions and concerns you raised in the original review are more constructive and additive to our technical contributions (rather than technical limitations) and hope that we have successfully addressed these points. If that's indeed the case, could you please consider raising your rating of the paper? If not, we would be happy to elaborate on any aspect you may still have questions. Thank you!

---

### Decision · Program_Chairs · 2023-09-21

**Decision:**

Accept (poster)

**Comment:**

This paper aims to improve the interpretability of SATNet by directly learning the matrix $C=S^T S$ rather than $S$ directly, transforming the learned matrix $C$ into a weighted MaxSAT formula, and then using an exact solver on the weighted MaxSAT formula during inference (optionally with additional domain knowledge incorporated). The reviewers felt that the proposed method was interesting and valuable in providing an interpretable representation of learned SATNet rules, that the intermediate experimental results were well-chosen to validate design choices, and that the evaluation on the 4 $\times$ 4 Sudoku experiment was convincing. The reviewers would have appreciated additional discussion on why the proposed method does not currently scale to 9 $\times$ 9 Sudokus (e.g., as mentioned in the reviewer discussion, by including the results on slow convergence and including more discussion on why the number of decoded clauses is so large).